# Genome-Wide Patterns of Homozygosity Reveal the Conservation Status in Five Italian Goat Populations

**DOI:** 10.3390/ani11061510

**Published:** 2021-05-23

**Authors:** Salvatore Mastrangelo, Rosalia Di Gerlando, Maria Teresa Sardina, Anna Maria Sutera, Angelo Moscarelli, Marco Tolone, Matteo Cortellari, Donata Marletta, Paola Crepaldi, Baldassare Portolano

**Affiliations:** 1Dipartimento Scienze Agrarie, Alimentari e Forestali, University of Palermo, Viale delle Scienze, 90128 Palermo, Italy; rosalia.digerlando@unipa.it (R.D.G.); mariateresa.sardina@unipa.it (M.T.S.); angelo.moscarelli@unipa.it (A.M.); marco.tolone@unipa.it (M.T.); baldassare.portolano@unipa.it (B.P.); 2Dipartimento Scienze Veterinarie, University of Messina, Polo Universitario dell’Annunziata, 98168 Messina, Italy; asutera@unime.it; 3Dipartimento di Scienze Agrarie ed Ambientali, Produzione, Territorio, Agroenergia, University of Milano, Via Celoria, 220133 Milano, Italy; matteo.cortellari@unimi.it (M.C.); paola.crepaldi@unimi.it (P.C.); 4Dipartimento di Agricoltura, Alimentazione, Ambiente, University of Catania, Via Valdisavoia, 95123 Catania, Italy; d.marletta@unict.it

**Keywords:** runs of homozygosity, inbreeding, local goat populations, genomic regions

## Abstract

**Simple Summary:**

In the local populations, the increase in inbreeding is a relevant problem for the reduction in production, reproduction, and adaptive traits. The application of genomic technologies has facilitated the assessment of inbreeding in these populations. The current study aims to investigate the patterns of homozygosity in five Italian local goat populations. The results showed the different selection histories and breeding schemes of these goat populations. The analysis also indicated the importance of this information to avoid future loss of diversity and to produce information for designing optimal breeding and conservation programs.

**Abstract:**

The application of genomic technologies has facilitated the assessment of genomic inbreeding based on single nucleotide polymorphisms (SNPs). In this study, we computed several runs of homozygosity (ROH) parameters to investigate the patterns of homozygosity using Illumina Goat SNP50 in five Italian local populations: Argentata dell’Etna (N = 48), Derivata di Siria (N = 32), Girgentana (N = 59), Maltese (N = 16) and Messinese (N = 22). The ROH results showed well-defined differences among the populations. A total of 3687 ROH segments >2 Mb were detected in the whole sample. The Argentata dell’Etna and Messinese were the populations with the lowest mean number of ROH and inbreeding coefficient values, which reflect admixture and gene flow. In the Girgentana, we identified an ROH pattern related with recent inbreeding that can endanger the viability of the breed due to reduced population size. The genomes of Derivata di Siria and Maltese breeds showed the presence of long ROH (>16 Mb) that could seriously impact the overall biological fitness of these breeds. Moreover, the results confirmed that ROH parameters are in agreement with the known demography of these populations and highlighted the different selection histories and breeding schemes of these goat populations. In the analysis of ROH islands, we detected harbored genes involved with important traits, such as for milk yield, reproduction, and immune response, and are consistent with the phenotypic traits of the studied goat populations. Finally, the results of this study can be used for implementing conservation programs for these local populations in order to avoid further loss of genetic diversity and to preserve the production and fitness traits. In view of this, the availability of genomic data is a fundamental resource.

## 1. Introduction

The application of genomic technologies, such as high-throughput genotyping, sequencing, and bioinformatics analysis has facilitated the assessment of genomic inbreeding based on single nucleotide polymorphisms (SNPs) as support to the traditional pedigree-based inbreeding coefficients [1,2]. Runs of homozygosity (ROH) are contiguous homozygous genotypes that are present in an individual since the identical haplotypes are inherited from each parent [3]. ROH have been used as predictors of inbreeding levels in livestock species [2,4,5,6,7]. In particular, a high frequency of long ROH may reflect recent inbreeding, while short ROH can be explained by the occurrence of ancient inbreeding [8]. Moreover, an increase in the homozygosity of certain regions of the genome may also take place as a result of natural or artificial selection, and shared ROH within populations produce ROH islands [9,10,11]. Previous studies showed that ROH islands can be used to identify genomic regions affecting the breed productive or adaptive traits and have been also able to reveal genetic and demographic events that contributed to model the population structure of these breeds [2,12,13,14,15], including for goat [16,17].

Nowadays, five native goat populations are reared in Sicily for milk production: Argentata dell’Etna, Derivata di Siria, Girgentana, Maltese, and Messinese. These populations present differences in both phenotypic and production traits. The Girgentana goat is one of the ancient Sicilian breeds, with long corkscrew horns in both sexes and a cream/white coat color; the Derivata di Siria breed (also known as Rossa Mediterranea) is completely red, whereas the Maltese, originating from the island of Malta, is white with a black head, and both breeds show long ears. Messinese and Argentata dell’Etna, due to their good adaptive traits and hardiness, are raised on farms located in marginal areas; the former is not officially recorded as a standard breed and presents a very different coat color, e.g., plain, pied or streaked, black, brown, or red with various shades; the latter has a coat color that has gray shading from light to dark, with silver glints, and gray skin. The breeding system is based on extensive or semi-extensive farming with a natural mating system [18]. In general, their current major breeding goal is to maintain their productive traits and maximize genetic diversity. Therefore, in these local populations, an investigation of genomic variation through estimates of inbreeding levels, is an important parameter to maintain their genetic distinctness and to ensure appropriate conservation [19].

The aim of this study was to investigate the patterns of homozygosity in these five local goat populations using medium density array. In addition, ROH islands were characterized with the aim of identifying genomic regions that harbor candidate genes that could potentially be associated to adaptive or productive traits.

## 2. Materials and Methods

### 2.1. Samples and Genotyping

A total of 177 goats, including Argentata dell’Etna (ARG, *n =* 48), Derivata di Siria (DDS, *n =* 32), Girgentana (GIR, *n =* 59), Maltese (MAL, *n =* 16), and Messinese (MES, *n =* 22), were randomly collected in different flocks located in different areas of Sicily. Genomic DNA was extracted from buffy coats of nucleated cells using a salting out method. The DNA samples were genotyped using the Illumina Goat SNP50 BeadChip. Markers were mapped using the goat assembly ARS1. The SNPs located on the X chromosome or with unknown position were discarded. The filters for quality control included the removal of markers with minor allele frequency (MAF) ≤0.02 and a call rate ≤0.98. Moreover, samples with more than 5% of missing genotypes were discarded from the analyses. The final working dataset retained for analyses included 174 animals and 48,348 markers.

### 2.2. Genetic Relationships and Population Structure

Multidimensional scaling analysis was performed using PLINK v 1.9 [20] based on pairwise identical-by-state (IBS) distances. The population structure was investigated by applying the model-based clustering algorithm run in ADMIXTURE from K = 2 to 5 [21]. The BITE R package was used to graphically represent the results [22].

### 2.3. Runs of Homozygosity Estimation

Runs of homozygosity (ROH) were detected using PLINK v 1.9 [20] with the following parameters: (i) minimum of 2 Mb in length, (ii) one missing SNP and one heterozygous genotype, (iii) the minimum number of SNPs for ROH was 20, (iv) the minimum SNP density was set to one SNP per 100 kb, with a maximum gap length of 500 kb. The mean number of ROH (MN_ROH_) and the average length of ROH (AL_ROH_) per animal were estimated. The ROH were classified in four groups based on the physical length: 2 to ≤4, 4 to ≤8, 8 to ≤16, and >16 Mb [23]. The mean sum of ROH for each group of each population was estimated by adding all ROH in that category and averaging per population. The inbreeding coefficient (F_ROH_) per individual was calculated as follows:

F_ROH_ = L_ROH_/Laut;

where L_ROH_ is the total length of ROH and Laut is the length of the autosomal genome (approx. 2450 Mb).

### 2.4. Runs of Homozygosity Islands

The shared genomic regions with occurrences of ROH (ROH islands) among the five goat populations were identified. The percentage of SNPs present in ROH was calculated based on the frequency of a SNP in them across individuals. The top 0.999 SNPs of the percentile distribution were selected as threshold in the meta-population. Information on the annotated genes within the ROH islands were obtained from the Genome Data Viewer tool provided by NCBI (https://www.ncbi.nlm.nih.gov/genome/gdv/browser/genome/?id=GCF_001704415.1, accessed 14 January 2021). Finally, we conducted a precise literature search.

## 3. Results

### 3.1. Genetic Relationships and Population Structure

The first dimension (C1) of multidimensional-scaling plot distinguished GIR from other populations (Figure 1a). Partial overlapping has been found between DDS and MAL, which also showed more spread clusters, while the ARG and MES individuals are positioned close to each other. The ADMIXTURE plots showed the results for K ranging from 2 to 5 (Figure 1b). In agreement with the results of C1 in the MDS plot, the first breed to be differentiated from the others was GIR (K = 2). Additional breed-specific cluster was observed at K = 3 for MAL and at K = 5 for DDS. Moreover, at this K value, ARG and MES exhibited the same genetic structure.

### 3.2. Runs of Homozygosity Detection

We detected a total of 3687 ROH >2 Mb across all autosomes, with an average of 22 ROH per individual. No ROH were identified in ten individuals, including seven ARG, two MES and one GIR. The descriptive statistics of ROH per populations are reported in Table 1. The mean number of ROH (MN_ROH_) ranged from 3.02 (ARG) to 38.81 (MAL). The average length of ROH (AL_ROH_) ranged from 4.98 Mb (ARG) to 9.61 Mb (DDS) and showed low variation (around 6.70 Mb) among GIR, MAL, and MES goat populations.

The majority of ROH segments (72%) were shorter than <8 Mb in length, while 374 segments (~10%) were longer than 16 Mb (Table 2). The longest ROH was 123.31 Mb, with 2491 SNPs and mapped on chromosome (*Capra hircus*) (CHI) 1 in the GIR breed.

To investigate the ROH patterns, the detected segments were clustered in four categories, and the average sum of ROH segment coverage per population was estimated (Figure 2). The highest mean ROH coverage in the short category (2–4 Mb) was found in GIR breed. The DDS and MAL goat breeds showed a larger mean portion of their genome (119.35 Mb and 110.175 Mb respectively) covered in longer ROH (>16 Mb). ARG and MES showed the lowest mean sum of ROH coverage in all the length categories.

The total number of ROH and the total genomic length for each animal of the five goat populations is shown in Figure 3. For ARG and MES, the majority of the animals clustered near to the origin of coordinates, carrying from 1 to 20 ROH with a total length <200 Mb. For the other three populations, there were individuals with a larger number of ROH (from 20 to 60) with a total length ranging from 200 to 600 Mb. In particular, GIR showed some animals with the total length of genome in ROH more than 600 Mb.

Then, we estimated the mean levels of genomic inbreeding (F_ROH_ > 2 Mb) for each population. MAL breed was characterized by the highest inbreeding (F_ROH_ = 0.125), followed by GIR (F_ROH_ = 0.108), whereas ARG showed the lowest value (F_ROH_ = 0.009).

### 3.3. Runs of Homozygosity Islands

Finally, we explored the genomic regions associated with ROH in the meta-population. Table 3 provides the position of each ROH island and the corresponding genes. In total, we detected three ROH islands (Table 3) on CHI05, CHI06, and CHI07, ranging in length from 1.97 Mb (36 consecutive SNPs on CHI07) to 3.42 Mb (20 consecutive SNPs on CHI06). Although the signals were moderate in height, the Manhattan plot showed few outstanding peaks with a highest occurrence of ROH (Figure 4).

Within these ROH islands, we identified 32 known genes together with uncharacterized loci (LOC) (Table 3). Only the ROH island on CHI07 did not harbor any genes.

## 4. Discussion

In the last few years, SNP arrays have been used to investigate the genomic background and ROH patterns in local or cosmopolitan goat breeds [4,10,23]. However, a deep characterization of ROH distribution in these five local goat populations remains unexplored. Therefore, using medium density SNPs array, we focused on the occurrence and distribution of ROH patterns. 

To understand the genetic relationship and the population structure, we carried out a MDS and admixture analysis (Figure 1); as expected, Girgentana was the most distinct breed, as reported in previous studies [18,24,25], due to its different origin area and isolated reproductive and breeding system. Argentata dell’Etna and Messinese individuals graphically are positioned close to each other (Figure 1) and share a substantial proportion of their ancestry. The genetic similarity between them could be explained considering their geographical and environmental closeness, management, and gene flow [25].

The ROH parameters (number, length, distribution) constitute an important source of information on livestock populations’ demographic history. Moreover, ROH could be used to infer inbreeding level for populations in which no pedigree records are available or where the records are of poor quality. In our study, the ROH parameters revealed well-defined differences among populations, as also shown in a previous study [4] in which the authors reported a complex case for goats from Mediterranean islands. They underlined that some populations displayed significant increased level of homozygosity while the others did not. Argentata dell’Etna and Messinese were the populations with the lowest mean number of ROH and inbreeding coefficient values (Table 1). Moreover, for these populations, a limited number of ROH (<20) per individual were identified, which covered less than 200 Mb (Figure 3), reflecting a low degree of recent inbreeding. For Argentata dell’Etna, the descriptive statistics here reported (Table 1) are lower than those observed by Cardoso et al. [4] (F_ROH_ = 0.02, AL_ROH_ = 47.01 and MN_ROH_ = 21.79). In the present study, we defined ROH as tracts of homozygous genotypes with length >2 Mb identified by a minimum number of 20 SNPs, as also reported in other studies in goat [23], whereas in Cardoso et al. [4], ROH were defined as homozygous genomic segments at least 1 Mb long, containing a minimum number of 15 SNPs. The use of different criteria for various parameters in the ROH identification (such as the minimum length, the minimum number of SNPs) between studies, as well as the differences in number of tested samples, could lead to these discrepancies in the descriptive statistics [26].

In general, AL_ROH_ values showed low variation among the goat populations, indicating that this value is not a good descriptor of ROH, as reported by other authors [27]. However, AL_ROH_ values detected in this study were comparable with those reported for local Spanish goat breeds (6.28 Mb) [28]. On the other side, the five goat populations presented different ROH length categories (Figure 2). In fact, clustering ROH into different length categories make it possible to detect and interpret genomic differences among breeds [7]. Derivata di Siria and Maltese breeds had a larger mean portion of their genome covered in long ROH (>16 Mb). Since the size of ROH is related to the inbreeding age, with longer ROH most likely due to more recent common ancestors and shorter ones due to older common ancestors [8], recent inbreeding can be supposed in Derivata di Siria and Maltese (Table 2). These data are important as long ROH are mostly enriched in region with deleterious mutations [29]. Pryce et al. [30], in an association study on genomic regions and inbreeding depression in Holstein cattle breed, reported that long ROH, such as those identified in our populations, was associated with reduction in milk yield. Therefore, these results also show the need of implementing conservation programs to maintain the local goat breeds in order to avoid further loss of genetic distinctiveness, and to preserve the production and fitness traits.

The differences among the five goat populations were also evident in the relationship between total number of ROH and total ROH coverage (Figure 3). For Girgentana and Maltese, there were several individuals that carried a large number of ROH and a large total length; in particular, in Girgentana some extreme individuals showed more than 600 Mb of their autosomes covered by ROH. The analysis of individual ROH may be useful for conservation programs in endangered populations, since animals with high levels of ROH and long segments, as observed in Girgentana, could be excluded or less used for mating purposes to minimize the loss in genetic diversity [31]. Moreover, these results reflect the higher ROH inbreeding coefficients estimated in the Maltese and Girgentana breeds compared to other three populations. However, none of the five populations showed high F_ROH_ values (>0.20). Bertolini et al. [17], in a study involving about 130 goat breeds, reported that ~60% of the breeds display low F_ROH_ (<0.10), also reported in our study for Argentata dell’Etna, Messinese, and Derivate di Siria, while ~30% and ~10% of the breeds showed moderate (0.10 < F_ROH_ < 0.20) or high (>0.20) F_ROH_ values, respectively. The results for F_ROH_ values also highlight the different selection histories and breeding schemes of the Sicilian goat populations. In particular, Argentata dell’Etna and Messinese are not subject to selection programs (with lowest F_ROH_ values), whereas Maltese and Derivata di Siria are characterized by limited selection program which, among other factors, can lead to an increase in inbreeding as also reported in cattle [5,32]. These results confirmed that ROH parameters are in agreement with the known demography of these populations, which experienced population bottlenecks in their history, accompanied by an inevitable increase in inbreeding.

The demography is an important factor that models the genomic patterns of homozygosity [33]. For example, the long segments detected in the Girgentata suggested that the inbreeding is more recent and indicative of demographic decline. In fact, the size of this population decreased almost 90% in 30 years as a consequence of the decrease in fresh goat milk consumption [34]. The breed is, indeed, as presented by the Food and Agriculture Organization, of endangered risk status [35], and about 1500 goats are currently being reared. Moreover, this breed is isolated from the other goat populations, as also confirmed in the MDS results (Figure 1a). Therefore, the ROH patterns results derive from the combined effects of small population size and the geographic isolation of Girgentana. Similar results were also found in other insular goat populations, such as those raised in Iceland or Madagascar, or in breeds that have suffered acute population declines due to competition with more productive foreign breeds and the progressive abandonment of low-income farming activities [17]. On the contrary, populations that allow gene flow from other populations had fewer and shorter ROH [26], as shown for Argentata dell’Etna and Messinese populations. These populations are reared in mountain areas under extensive and semi-extensive systems, and clear evidence of gene flow between them has been reported [25]. Gene flows would result in a breaking down of ROH, particularly reducing the length of the segments in these populations as also shown by Bertolini et al. [17], in which the total ROH length was reported to be much shorter in crossbred goats. The results also confirmed that Girgentana has not recently been crossed with other breeds [36], otherwise the long ROH would have broken down. Moreover, the prevalence of long ROH in these populations, is consistent with the limits of an effective genetic management resulting from the pedigree data of poor quality. Therefore, the analysis of ROH also highlights the importance of novel marker-based information to avoid future loss of diversity. 

Shared ROH in the meta-population identify common genomic regions in which reduced haplotype variability produces ROH islands [9]. Consequently, high homozygosity around the selected locus might harbor targets of positive selection. To identify genomic regions with the highest frequency of ROH and thus containing candidate genes potentially under selection, we examined ROH islands with several candidate genes that could potentially be associated to adaptive or productive traits. In our study, we found three ROH islands with several known genes together with uncharacterized genes (located on chromosome (LOC)). The ROH island on CHI05 (35.75–38.31 Mb) matched with a ROH signal in the Pyrenean goat population and with ROH and positive selection (iHS) signals in goat breeds of the northwestern Africa [10]. *ADAMTS20* is one of the most known genes within this genomic region. This gene was found to be involved in coat color variation during the event of goat domestication and associated with melanocyte development [37]. Liu et al. [38] reported that *ADAMTS20* gene presented copy number variation in goats, with high frequency in the Eastern Mediterranean goat breeds. Di Gerlando et al. [39] showed that this gene is also within a CNV in the same populations here investigated. Moreover, this genomic region showed four genes (*PRICKLE1*, *ZCRB1*, *YAF2*, and *GXYLT1*) reported to be associated with paratuberculosis resistance in cattle [40]. The ROH island on CHI06 (35.35–38.77 Mb) overlapped with selection signatures identified in Egyptian local goats for grazing stress tolerance [41]. This ROH island harbored several genes previously reported to be strongly linked with positive selection in other domestic livestock. These genes included *ABCG2*, *SPP1*, *MEPE*, *LAP3*, and *MED28* associated with milk production traits [42,43] and *NCAPG*, *IBSP*, *DCAF16*, *FAM184B*, and *LCORL* involved in feed intake in livestock [44,45]. Other genes are relevant to estrogen, follicular growth, and litter size (SPP1) [46] and in immune response (*HERC5*, *HERC3*) [47]. All these genes, related with milk production, reproduction, and immune response, are consistent with the phenotypic traits of the studied goat populations. In fact, these populations, historically farmed in Sicily, possess valuable traits such as disease resistance, high fertility, and adaptation to harsh conditions, representing an important reservoir of diversity that may turn out useful to face the upcoming climate change. Moreover, some of these genomic regions overlapped with previously reported ROH islands or selection signatures, providing evidence that they are not artifacts but genomic regions affected by high homozygosity across all analyzed populations. Therefore, these aspects indicate that the individuals under investigation may have experienced selective pressure on their genomes for these specific traits, which has contributed to the formation of these ROH islands.

## 5. Conclusions

In this study, we investigated the occurrence and distribution of ROH patterns using medium density SNPs array in five local goat populations. The ROH parameters revealed well-defined differences and are in agreement with the known demography of these populations as well as with respect to their breeding history and population size. For Argentata dell’Etna and Messinese, the ROH patterns showed a low degree of recent inbreeding and gene flow between them. The long segments detected in the Derivata di Siria and Maltese could seriously impact the overall biological fitness of these breeds since that long ROH can be enriched in genomic regions that carry deleterious mutations. In Girgentana, the ROH patterns are the result of the combined effects of small population size and geographic isolation, which can endanger the viability of the breed. The description of ROH islands evidenced the effects of the combination of genetic events and selection history on the genome of these important local genetic resources. The analysis of ROH highlights the importance of novel marker-based information to prevent future loss of diversity and to produce information for designing optimal breeding and conservation programs.

## Figures and Tables

**Figure 1 animals-11-01510-f001:**
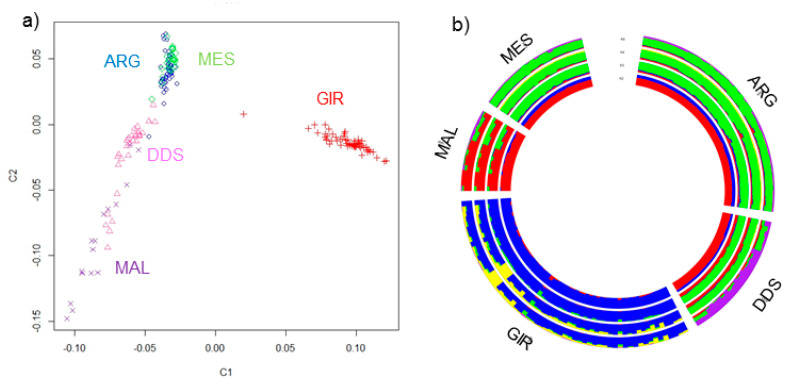
(**a**) Multidimensional scaling (MDS) analysis of the five goat populations. The first (C1) and second (C2) components are presented in the x- and the y-axis, respectively. (**b**) Maximum likelihood estimation calculated with the admixture algorithm. The inferred clusters (K) were represented from K = 2 to 5. Argentata dell’Etna (ARG), Derivata di Siria (DDS), Girgentana (GIR), Maltese (MAL), and Messinese (MES).

**Figure 2 animals-11-01510-f002:**
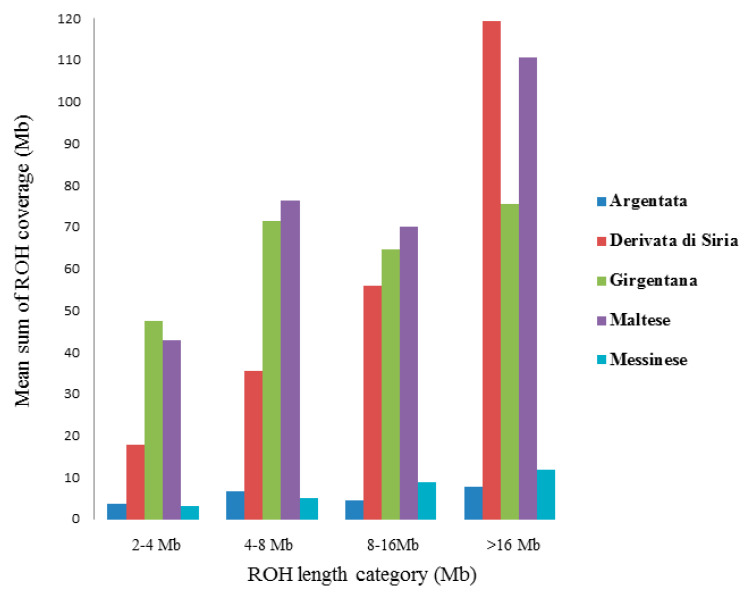
Classification of runs of homozygosity (ROH) in four categories according to size (x-axis) and mean sum of ROH in Mb (y-axis) within each ROH length category per population.

**Figure 3 animals-11-01510-f003:**
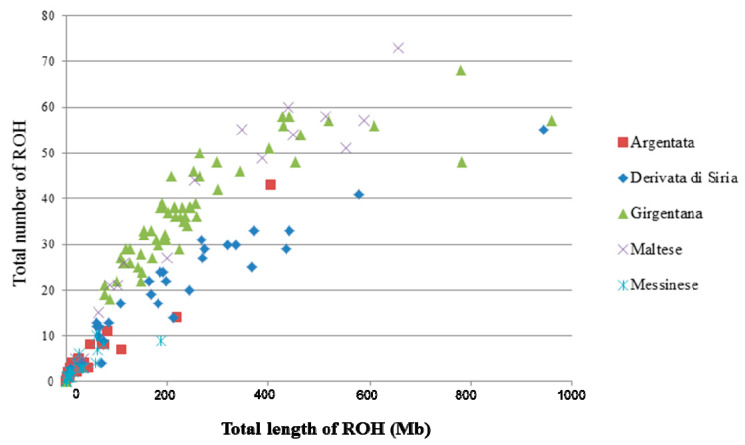
Relationship between the total number of runs of homozygosity (ROH) segments (y-axis) and the total length (Mb) of genome in such ROH (x-axis) for each individual of the five goat populations.

**Figure 4 animals-11-01510-f004:**
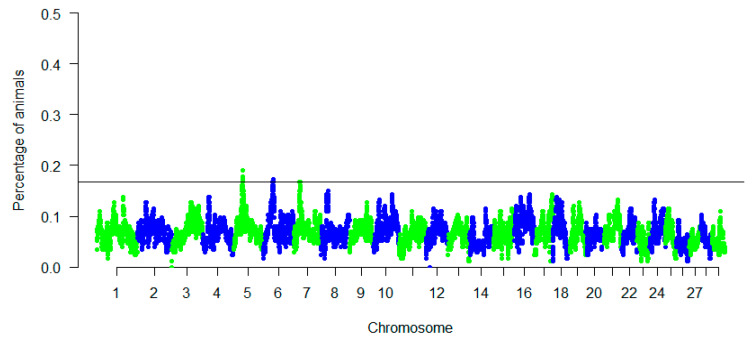
Manhattan plot of the incidence of each SNP in the runs of homozygosity among all goat populations. A threshold of 0.16 was chosen to detect the regions.

**Table 1 animals-11-01510-t001:** Summary statistics of runs of homozygosity (ROH) and inbreeding coefficients in the five goat populations. Argentata dell’Etna (ARG), Derivata di Siria (DDS), Girgentana (GIR), Maltese (MAL), and Messinese (MES).

Population	F_ROH_ > 2Mb ± s.d ^1^	N ^2^	AL_ROH_ ^3^	TN_ROH_ ^4^	MN_ROH_ ^5^
ARG	0.009 ± 0.016	7	4.98 Mb	139	3.0
DDS	0.097 ± 0.083	0	9.61 Mb	667	21.5
GIR	0.108 ± 0.076	1	6.42 Mb	2190	37.1
MAL	0.125 ± 0.094	0	6.86 Mb	621	38.8
MES	0.012 ± 0.018	2	6.70 Mb	70	3.2

^1^ F_ROH_ = Mean ROH-based inbreeding coefficient with standard deviation (s.d); ^2^ N = number of individuals from the samples for which no ROH were detected; ^3^ AL_ROH_, average length of ROH per individual and per population in Mb; ^4^ TN_ROH_, total number of ROH per population; ^5^ MN_ROH_, mean number of ROH per individual.

**Table 2 animals-11-01510-t002:** Descriptive statistics of the number (N_ROH_) and the frequency (Freq) distribution of runs of homozygosity (ROH) in four ROH length categories (Mb) in the five goat populations. Argentata dell’Etna (ARG), Derivata di Siria (DDS), Girgentana (GIR), Maltese (MAL), and Messinese (MES).

Length (Mb)	ARG		DDS		GIR		MAL		MES	
	N_ROH_	Freq	N_ROH_	Freq	N_ROH_	Freq	N_ROH_	Freq	N_ROH_	Freq
2–4	56	0.40	183	0.27	923	0.42	223	0.36	23	0.33
4–8	51	0.37	205	0.31	764	0.35	222	0.36	21	0.30
8–16	19	0.14	150	0.22	349	0.16	107	0.17	17	0.24
>16	13	0.09	129	0.20	154	0.07	69	0.11	9	0.13

**Table 3 animals-11-01510-t003:** Genomic regions with the highest frequency of runs of homozygosity (ROH islands) occurrence across all individuals.

Chr ^1^	Start	End	Genes
5	35752670	38314749	*TMEM17, TWF1, IRAK4, PUS7L, ADAMTS20, PRICKLE1, ZCRB1, YAF2, GXYLT1*
6	35353261	38772649	*SNCA, GPRIN3, TIGD2, FAM13A, HERC3, NAP1L5, PIGY, HERC5, HERC6, PPM1K, ABCG2, PDK2, SPP1, MEPE, IBSP, TRNAA-CGC, LAP3, MED28, FAM184B, DCAF16, NCAPG, LCORL, TRNAC-GCA*
7	24117620	26089250	*LOC108633170, LOC102170513, LOC102170229*

^1^ Chr: *Capra hircus* chromosome.

## Data Availability

The data that support the findings of this study are available on request from the corresponding author.

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
