# Peer review of "Genome-Wide Patterns of Homozygosity Reveal the Conservation Status in Five Italian Goat Populations"

_animals, 2021, doi:10.3390/ani11061510_

Round 1
Reviewer 1 Report
The manuscript provides an interesting work to detect inbreeding levels in local Italian goat breeds. Several sections need significant improvement as highlighted in the comments.
Title: Change “local goat populations” to “five Italian goat breeds” or “five Sicilian dairy goat breeds”, or with something similar to include a regional name identity.
Line 28: Please elaborate “local”, e.g., Italian local (native?) breeds.
Provide sample size and origin of samples in abstract.
Line 32: What is Girgentana, a breed or a population? Population and breed are not the same. Please clarify and be consistent throughout the manuscript.
Line 38: What are the main insights? Report in the abstract.
Introduce the major differences between the 5 breeds/populations.
Line 85: It is PLINK v1.07 (not 1.7), which is quite old and may have used v1.9.
I strongly suggest to do a STRUCTURE/Admixture analyses to see the Gene flow and across bread relatedness.
Lines 101, 164: What is “meta-population”?
Figure 1: What is on x-axis and y-axis?
Lines 180-185, 192-195: Repetition of intro.
Line 306: Change “contribute” to “contributed”.
Lines 308-317: Conclusions are also much generalized, same as abstract. Both sections need to highlight the major findings or insights, as key messages.
Author Response
REVIEWER 1
The manuscript provides an interesting work to detect inbreeding levels in local Italian goat breeds. Several sections need significant improvement as highlighted in the comments.
RESPONSE: Thanks for the general comment and for the suggestions.
Title: Change “local goat populations” to “five Italian goat breeds” or “five Sicilian dairy goat breeds”, or with something similar to include a regional name identity.
RESPONSE: Thanks for the suggestion. The title has been revised. Lines 2-3
Line 28: Please elaborate “local”, e.g., Italian local (native?) breeds.
RESPONSE: Done. Line 27
Provide sample size and origin of samples in abstract.
RESPONSE: Done. Lines 27-28
Line 32: What is Girgentana, a breed or a population? Population and breed are not the same. Please clarify and be consistent throughout the manuscript.
RESPONSE: Thanks for the comment. Argentata dell’Etna, Derivata di Siria, Girgentana and Maltese are officially recognized breeds. The Messinese goat is not officially recorded as a standard breed like many other goat populations. This is added in the revised version. In the manuscript, we have chosen to use the term populations when we refer to all five goats at the same time.
Line 38: What are the main insights? Report in the abstract.
RESPONSE: Thanks for the suggestion. The sentence has been modified. Lines 35-37
Introduce the major differences between the 5 breeds/populations.
RESPONSE: Thanks for the suggestion. The information on the major differences among populations have been reported in the introduction. Lines 61-71.
Moreover, the differences are also reported in the discussion section.
Line 85: It is PLINK v1.07 (not 1.7), which is quite old and may have used v1.9.
RESPONSE: Done. Lines 94 and 99.
I strongly suggest to do a STRUCTURE/Admixture analyses to see the Gene flow and across bread relatedness.
RESPONSE: Thanks for the suggestion. The required analysis (Fig 1b in the revised version) has been added in the manuscript. Lines 95-97; 125-129; 133-135; 212, 216-217.
Lines 101, 164: What is “meta-population”?
RESPONSE: A meta-population is defined as a set of potential local populations with the small size and limited geographic distribution. In this manuscript, we used the meta-population when the five goat populations are considered as a single group.
Figure 1: What is on x-axis and y-axis?
RESPONSE: The figure 1 and the title have been revised. Lines 132-133.
Lines 180-185, 192-195: Repetition of intro.
RESPONSE: Thanks for the comment. We used these sentences to introduce the discussion of the obtained results. However, the sentences have been partially modified.
Line 306: Change “contribute” to “contributed”.
RESPONSE: Done
Lines 308-317: Conclusions are also much generalized, same as abstract. Both sections need to highlight the major findings or insights, as key messages.
RESPONSE: Thanks for the suggestion. Both sections have been improved in the revised manuscript.
Reviewer 2 Report
Authors investigated the occurrence and distribution of ROH patterns using 308 medium density SNPs array in five local goat populations. The results were obtained from a consistent sample, and the obtained results may be of interest for the field. Several points need to be considered before publication.
1.L19: What exactly are phenotypic values? Please list specific phenotypes.
2.L74: Is the population a hybrid population? If so, how many generations is the population?
3.L82:It’s PLINK v1.07, not PLINK v1.7. Please correct the following similar errors.
4.L86: Why set the minimum length to 2Mb? What is the reference?
- Figure 1 lacks coordinate names. Please add them.
- The clarity of Figure 1 is not enough, please modify it. The same problem appears in Figure 2 and 3.
- What are the criteria for setting the threshold in Figure 4.
Author Response
REVIEWER 2
Authors investigated the occurrence and distribution of ROH patterns using medium density SNPs array in five local goat populations. The results were obtained from a consistent sample, and the obtained results may be of interest for the field. Several points need to be considered before publication.
RESPONSE: Thanks for the general comment and for the suggestions.
1.L19: What exactly are phenotypic values? Please list specific phenotypes.
RESPONSE: Thanks for the comment. We changed “phenotypic values” to “production, reproduction and adaptive traits”. Line 18
2.L74: Is the population a hybrid population? If so, how many generations is the population?
RESPONSE: No, the Derivata di Siria (also known as Rossa Mediterranea) is the name of an old local dairy goat breed reared in Sicily.
3.L82:It’s PLINK v1.07, not PLINK v1.7. Please correct the following similar errors.
RESPONSE: Thanks; however, we realized that the version was wrong; in the revised manuscript, we reported the correct version: PLINK 1.9. Lines 94 and 99.
4.L86: Why set the minimum length to 2Mb? What is the reference?
RESPONSE: In our study, we did not perform LD pruning, but, owing to the minimum 2Mb size of ROH segments, we tried to avoid small autozygous segments caused by LD. The same minimum length has been reported in a study of local goat (Onzima et al., 2018 - Frontiers in Genetics 9:318). The reference is reported in the discussion section.
- 5. Figure 1 lacks coordinate names. Please add them.
RESPONSE: Thanks. The figure 1 and the title have been revised. Lines 132-133
- The clarity of Figure 1 is not enough, please modify it. The same problem appears in Figure 2 and 3.
RESPONSE: The Figures have been modified in the revised manuscript and the clarity have been improved.
- What are the criteria for setting the threshold in Figure 4.
RESPONSE: Thanks for the comment. Given that the definition of ROH parameters are still not unambiguous, we used the same threshold reported in recent studies (e.g. Schiavo et al., 2020; Moscarelli et al., 2021).
Reviewer 3 Report
The manuscript described the genome-wide characteristics of ROHs in five local goat populations, and the results revealed the breed differences and the different selection histories. Local livestock breeds are very important nature resources, any studies for revealing germplasm characteristics are of importance. For this investigation, it is meaningful for future guiding the conservation, breeding and production for these local goat breeds. Before acceptance, some comments should be addressed below.
- Because five native goat populations are reared in Sicily for milk production in history, in the annotations of genes harbored by detectable ROHs, the authors should give highlighted issues in discussion part of the manuscript.
- As personal suggestion, in the discussion, the authors should give more theoretical discussions or further considerations on those breed-specific SNPs that are not designed on the commercial Illumina Goat SNP50. If not rational, the author can directly refuse.
- Compared with continent or mainland, the area of Sicily island is much smaller, the gene interflow between breeds may be very easy, the reviewer will be glad to read that what or how produce the breed differences (e.g., inbreed degree) in the history, or give a figure that combined a Sicily map with five breeds’ locations.
- When quality control for the SNPs, why the authors used MAF ≤ 0.02? It is not common compared to the cutoff of 0.01 or 0.05.
- The result of 3.1 only has one sentence, and it is not common in normal expression in paper, merge it with the other result parts.
- Abstract must be absolutely clear. Line 34, a full stop or period is needed.
Author Response
REVIEWER 3
The manuscript described the genome-wide characteristics of ROHs in five local goat populations, and the results revealed the breed differences and the different selection histories. Local livestock breeds are very important nature resources, any studies for revealing germplasm characteristics are of importance. For this investigation, it is meaningful for future guiding the conservation, breeding and production for these local goat breeds. Before acceptance, some comments should be addressed below.
RESPONSE: Thanks for the general comment and for the suggestions.
- Because five native goat populations are reared in Sicily for milk production in history, in the annotations of genes harbored by detectable ROHs, the authors should give highlighted issues in discussion part of the manuscript.
RESPONSE: Thanks for the suggestion. A sentence has been added in the discussion section. Lines 330-333.
- As personal suggestion, in the discussion, the authors should give more theoretical discussions or further considerations on those breed-specific SNPs that are not designed on the commercial Illumina Goat SNP50. If not rational, the author can directly refuse.
RESPONSE: Thanks for the suggestion. Considering the general aim of the present study, with the identification of shared ROH islands in the five goat populations, the request on breed-specific SNPs is out from the goal of the work.
- Compared with continent or mainland, the area of Sicily island is much smaller, the gene interflow between breeds may be very easy, the reviewer will be glad to read that what or how produce the breed differences (e.g., inbreed degree) in the history, or give a figure that combined a Sicily map with five breeds’ locations.
RESPONSE: Thanks for the suggestion. We added the Admixture analysis to understand the genetic structure and gene flow of these populations. Moreover, in the text we reported the possible motivations which produce the breed differences. For example, Girgentana was the most distinct breed, due to different origin area, and isolated reproductive and breeding system. The genetic similarity between Argentata dell’Etna and Messinese could be explained considering their geographical and environmental closeness, management, and shared ancestral components. E.g. Lines 212-219; 271-278; 285-287; 293-295
- When quality control for the SNPs, why the authors used MAF ≤ 0.02? It is not common compared to the cutoff of 0.01 or 0.05.
RESPONSE: Thanks for the comment. There is no consensus in the literature about the best parameters for MAF in quality control for estimation of ROH parameters, and most of the publications reported a range of 0.01-0.05, as well as suggested by the reviewer. In this manuscript, we used the same MAF value (0.02) reported in a recent study on Italian local pig populations (Schiavo et al., 2020), and within the range reported in the literature.
- The result of 3.1 only has one sentence, and it is not common in normal expression in paper, merge it with the other result parts.
RESPONSE: Thanks for the suggestion. Done
- Abstract must be absolutely clear. Line 34, a full stop or period is needed.
RESPONSE: Thanks for the suggestion. We modified the text of the abstract adding more information.
Reviewer 4 Report
To my opinion the manuscript is well written, except for some minor changes to make as in the detailed review.

Author Response
REVIEWER 4
Line 22: done
Line 34: done
Lines 208-210. For Argentata dell’Etna, the descriptive statistics reported in this study are lower than those observed by Cardoso et al. In our study, we defined ROH as tracts of homozygous genotypes with length >2 Mb identified by a minimum number of 20 SNPs, whereas in Cardoso et al., ROH were defined as homozygous genomic segments at least 1 Mb long, containing a minimum number of 15 SNPs. The use of these different criteria for ROH detection and differences in number of tested samples could lead to these discrepancies in the descriptive statistics between studies. The sentence has been revised. Lines 236-239
Lines 212-213 The sentence has been revised. Lines 240-241
Line 214: done
Line 245: done
Line 306: done
Reviewer 5 Report
Mastrangelo et al. (animals-1188601-peer-review-v1) computed several runs of homozygosity (ROH) parameters to investigate the patterns of homozygosity using Illumina Goat SNP50 in five local populations. The analysis of ROH highlights the importance of novel marker-based information to prevent future loss of diversity and to produce information for designing optimal breeding and conservation. The result is important for goat industry and the manuscript is generally well written. The simple summary and abstract are informative. Introduction is sufficient to explain the reasons why the study was conducted. Materials and methods are appropriate and adequately described. Conclusion are supported by the presented data.
I have few minor comments for improvement of the study.
①Title: Maybe the paper contents are not enough to support the title, it is better to exchange a new one.
②Line 33 “The Derivata di Siria and Maltese breeds showed the presence in their genome of long ROH (>16 Mb),” There should be a comma after this sentence.
③Line 74 It is better to specify sample collection and DNA extraction method.
④Line 76 It is better to add the website after “Markers were mapped using the goat assembly ARS1.”
⑤Line 106 In the results section, it is better to place the agarose gel electrophoresis of genomic DNA.
⑥Line 107 “Samples and Genotyping”, to make it clear, it is better to list the SNP quality control statistic as a table
⑦All diagrams should be self-explanatory, so graph and table notes need to be detailed.
⑧The English writing needs to be polished.
Author Response
REVIEWER 5
Mastrangelo et al. (animals-1188601-peer-review-v1) computed several runs of homozygosity (ROH) parameters to investigate the patterns of homozygosity using Illumina Goat SNP50 in five local populations. The analysis of ROH highlights the importance of novel marker-based information to prevent future loss of diversity and to produce information for designing optimal breeding and conservation. The result is important for goat industry and the manuscript is generally well written. The simple summary and abstract are informative. Introduction is sufficient to explain the reasons why the study was conducted. Materials and methods are appropriate and adequately described. Conclusion are supported by the presented data.
I have few minor comments for improvement of the study.
RESPONSE: Thanks for the general comment and for the suggestions.
①Title: Maybe the paper contents are not enough to support the title, it is better to exchange a new one.
RESPONSE: Thanks for the suggestion. The title has been revised. Lines 2-3
②Line 33 “The Derivata di Siria and Maltese breeds showed the presence in their genome of long ROH (>16 Mb),” There should be a comma after this sentence.
RESPONSE: Done
③Line 74 It is better to specify sample collection and DNA extraction method.
RESPONSE: Thanks for the suggestion. The information has been added in the revised manuscript. Lines 84-86
④Line 76 It is better to add the website after “Markers were mapped using the goat assembly ARS1.”
RESPONSE: we did not use information from websites to update the map file.
⑤Line 106 In the results section, it is better to place the agarose gel electrophoresis of genomic DNA.
RESPONSE: The concentration and the quality of extracted DNA was checked using NanoDrop ND-1000 spectrophotometer.
⑥Line 107 “Samples and Genotyping”, to make it clear, it is better to list the SNP quality control statistic as a table
RESPONSE: We took in examination the reviewer’ suggestion to list the SNP quality control statistic as a table. However, we believe it is unnecessary to insert a table to list the SNP quality control statistic, since this is well described in the text of Materials and Methods: “The filters for quality control included the removal of markers with minor allele frequency (MAF) ≤ 0.02 and a call rate ≤ 0.98. Moreover, samples with more than 5% of missing genotypes were discarded from the analyses”.
⑦All diagrams should be self-explanatory, so graph and table notes need to be detailed.
RESPONSE: Thanks for the suggestion. We checked and improved the notes in tables and figures in the revised version.
⑧The English writing needs to be polished.
RESPONSE: Thanks for the suggestion. The English has been revised.
Round 2
Reviewer 1 Report
Well done for significantly improving the manuscript. A minor suggestion is to improve the quality of Figures. Please also check Figure 4, and confirm if the Y-axis label is correct. It might be "Proportion of animals".